# Ceramides—Emerging Biomarkers of Lipotoxicity in Obesity, Diabetes, Cardiovascular Diseases, and Inflammation

**DOI:** 10.3390/diseases12090195

**Published:** 2024-08-23

**Authors:** Ginka Delcheva, Katya Stefanova, Teodora Stankova

**Affiliations:** Department of Medical Biochemistry, Faculty of Pharmacy, Medical University of Plovdiv, 4002 Plovdiv, Bulgaria; katya.stefanova@mu-plovdiv.bg (K.S.); teodora.stankova@mu-plovdiv.bg (T.S.)

**Keywords:** ceramides, lipotoxicity, diabetes, cardiovascular diseases, inflammation

## Abstract

Abnormalities in lipid homeostasis have been associated with many human diseases, and the interrelation between lipotoxicity and cellular dysfunction has received significant attention in the past two decades. Ceramides (Cers) are bioactive lipid molecules that serve as precursors of all complex sphingolipids. Besides their function as structural components in cell and mitochondrial membranes, Cers play a significant role as key mediators in cell metabolism and are involved in numerous cellular processes, such as proliferation, differentiation, inflammation, and induction of apoptosis. The accumulation of various ceramides in tissues causes metabolic and cellular disturbances. Recent studies suggest that Cer lipotoxicity has an important role in obesity, metabolic syndrome, type 2 diabetes, atherosclerosis, and cardiovascular diseases (CVDs). In humans, elevated plasma ceramide levels are associated with insulin resistance and impaired cardiovascular and metabolic health. In this review, we summarize the role of ceramides as key mediators of lipotoxicity in obesity, diabetes, cardiovascular diseases, and inflammation and their potential as a promising diagnostic tool.

## 1. Introduction

Abnormalities in lipid homeostasis have been associated with many human diseases, leading to the rise of lipidomics as an emerging field studying lipids and factors that interact with lipids [1,2]. The term “lipotoxicity” was first adopted in 1994 by Lee et al. in the context of the pathogenesis of obesity-related β-cell alterations. The study described how lipid overload in pancreatic β-cells led to loss of function and the onset of type 2 diabetes mellitus (T2DM) in rats [3]. Lipotoxicity refers to the process of excess accumulation of lipids in non-adipose tissues and over-activation of lipid signaling pathways that induce cellular dysfunction. The increased concentrations of lipids and lipid derivatives result in many simultaneously occurring metabolic and functional disturbances in multiple organs and systems. The main organs affected include the pancreas, liver, skeletal muscles, heart, and kidneys [4]. The interrelation between lipotoxicity and cellular dysfunction has focused significant attention in the past two decades on a wide range of prevalent diseases, including obesity, diabetes, CVDs, cancer, and neurodegenerative diseases [4,5,6,7].

Ceramides (Cers) are bioactive lipid molecules that serve as precursors of all complex sphingolipids. Ceramides are composed of sphingosine acylated with a fatty acid, usually saturated or monounsaturated, with a chain length ranging from 14 to 26 carbon [8]. Besides their function as structural components in cell and mitochondrial membranes, Cers play significant role as key mediators in cell metabolism and are involved in numerous cellular processes, such as proliferation, differentiation, inflammation, and induction of apoptosis. Ceramides are considered potential lipotoxic agents that modulate various processes and cause metabolic dysfunction [9,10,11,12]. Some of the effects of ceramide accumulation on several pathways include inhibition of PI3K/Akt pathway (impairing insulin signaling and glucose uptake), inhibition of hormone-sensitive lipase (directly leading to excessive storage of triglycerides), overproduction of ROS in mitochondria via the inhibition of complex III and IV of the respiratory chain (reducing intracellular ATP production and causing oxidative stress and apoptosis), and activation of proinflammatory mediator synthesis, such as TNF-α, interferon-γ, and IL-1β (inducing inflammation) [8,9,10,11,12,13].

## 2. Ceramide Synthesis and Transport

The synthesis and degradation of ceramides play crucial roles in maintaining cell homeostasis. Ceramides, the hub of the sphingolipid pathway, can be produced through three different pathways: (1) de novo synthesis, (2) sphingomyelin hydrolysis, or (3) salvage pathway (Figure 1). De novo synthesis occurs in the endoplasmic reticulum and starts with the condensation of serine with palmitoyl-CoA to form 3-ketosphinganine at the cytosolic surface of the ER, which is the rate-limiting step that is catalyzed by serine palmitoyltransferase. The carbonyl group of 3-ketosphinganine is then reduced by 3-ketosphinganine reductase to form sphinganine, which can be acylated with a fatty acyl-CoA by one of six (dihydro)ceramide synthases to form dihydroceramide. Dihydroceramide desaturase inserts a double bond into the sphingoid base to form ceramides. Once synthesized, Cers are transported into the Golgi apparatus via ceramide transfer protein or transport vesicles, where they can be converted to complex sphingolipids. In the second pathway, Cers are synthesized via the hydrolysis of the phosphocholine head groups of sphingomyelin (SM) by five different isoforms of sphingomyelinases. Because the amount of SM is much higher than the amount of total ceramide in the cell, hydrolysis of a small portion of SM significantly changes ceramide levels. In the salvage pathway, Cers are recycled from sphingosine re-acylation that is catalyzed by ceramide synthase [8,11,12,13,14,15,16,17].

Twenty years ago, ceramides were thought to be a homogenous class of sphingolipids, but now it is known that ceramides exert fundamentally different biological roles depending on the fatty acyl chain lengths [6]. Ceramides are largely synthesized by six different fatty acyl selective ceramide synthase (CerS) isoforms (CerS1-CerS6) with specific tissue distribution. Each CerS exhibits substrate specificity toward acyl-CoAs carrying different chain lengths (CerS1 is mainly expressed in muscles, C18; CerS2 is mainly expressed in liver, kidney, small intestine, and heart, C22 and C24; CerS4, C20, C22, C24, CerS5, C16, and CerS6 are mainly expressed in adipose tissue, C16; and only CerS3 exhibits broad substrate specificity toward medium- to long-chain fatty acyl-CoAs) (Table 1) [11,12,17,18,19,20,21,22,23]. In humans, CerS enzymes are involved in the regulation of various cellular and metabolic functions. For example, C18:0 ceramide produced by CerS1 is essential for brain development and neuronal signaling, while ceramides produced by CerS2 maintain the normal functions of liver, lungs, brain, heart, and kidney. However, it has been found that dysregulation of CerS leads to increased levels of various ceramide subspecies that have a key role in the pathophysiology of various human diseases [12].

In plasma, ceramides are carried by low-density and high-density lipoproteins (LDL and HDL). LDL is the major carrier of Cers in the plasma (about 60% of total plasma Cers), whereas HDL carries about 24% of plasma Cers. In addition, Cers are also found in extracellular vesicles (lipid vesicles secreted by cells) [24].

The accumulation of various ceramides in tissues causes metabolic and cellular dysfunction [25]. In recent years, Cers have received attention for their prominent role in obesity, metabolic syndrome, T2DM, atherosclerosis, and cardiovascular diseases (CVDs) [6,8,10,11,13,26,27]. Plasma levels of ceramides are increased in patients with obesity and type 2 diabetes mellitus, and are associated with insulin resistance and impaired cardiovascular and metabolic health [10,27,28,29,30,31].

## 3. Ceramides and Obesity

Overweight and obesity are global health problems, and their prevalence is constantly increasing [32]. According to a WHO report, the worldwide prevalence of obesity has tripled since 1975, mainly due to sedentary lifestyle and unhealthy diets [33]. It is estimated that more than 2.1 billion people worldwide (30% of the global population) are obese or overweight, and a total of 380 million children and adolescents are affected [34,35]. Obesity is characterized by excessive accumulation of body fat within adipose tissue, which has detrimental effects on human health [36]. Body mass index (BMI) is the most commonly used measure of overweight and obesity in men and women and is calculated by dividing weight in kilograms by height in meters squared. A BMI between 25 and 29.9 indicates overweight, and a BMI ≥ 30 indicates obesity.

Obesity is a major risk factor for the development of a number of chronic and life-threatening diseases, including dyslipidemia, insulin resistance, type 2 diabetes, and CVDs [8,25,26]. Excess calorie intake results in dyslipidemia and increased fatty acid supply, which leads to accumulation of lipid metabolites in peripheral tissues such as the liver, skeletal muscle, heart, and pancreas. The biologically active lipids that accumulate in obesity include triacylglycerol (TAG), diacylglycerol (DAG), long-chain fatty acids, acylcarnitine, phospholipids, and ceramides [6,25,37,38]. It is reported that ceramides are increased in the skeletal muscle, liver, and hypothalamus of obese rodents and humans. Different mechanisms lead to the increased formation of ceramides in peripheral tissues in obesity and lipid excess [25,39]. One of these is increased plasma-free fatty acids used for de novo ceramide synthesis and another is the increased expression of a number of genes involved in de novo synthesis. Other factors that may impair ceramide metabolism in obesity include pro-inflammatory cytokines, oxidative stress, and hormones.

Previous studies report the role of adipokine adiponectin in the control of ceramide levels [8,10]. Adiponectin is predominantly secreted from mature white adipocytes and regulates glucose and lipid homeostasis acting on many tissues, including the liver, kidney, adipose tissue, and pancreas, and exerting anti-diabetic, anti-inflammatory, and cardioprotective actions [8,10,30]. Adiponectin receptors AdipoR1 and AdipoR2 possess ceramidase activity that is activated by adiponectin binding and degrades ceramide into sphingosine, decreasing ceramide levels [6,8,10,20,30,37,40,41]. Sphingosine can be converted to sphingosine-1-phosphate (S1P) by sphingosine kinase (SphKs) [19]. Two major isoforms of SphKs in mammals have been described, namely, SphK1 and SphK2. S1P is a bioactive lipid signaling molecule that controls many physiological and pathological processes by binding with five different G-protein-coupled S1P receptors. The concentration of S1P is higher in plasma than in tissues and is carried mainly by HDL and albumin [42].

The reciprocal balance of ceramide and S1P has been termed ceramide/S1P rheostat. These two sphingolipid metabolites are important second messengers able to regulate cellular functions by modulating opposing signaling pathways. Ceramide has been shown to induce apoptosis, whereas S1P stimulates cell survival, proliferation, and tissue regeneration. Hence, maintaining the ceramide/sphingosine-1-phosphate balance is crucial for cells and cell fate decisions (Figure 2) [43,44,45]. Because ceramide and S1P are interconvertible metabolites, an increase in ceramide levels causes a decrease in S1P and vice versa. Therefore, the ceramide/S1P rheostat has been explored for the development of new therapeutic strategies for various diseases [46].

Sphingosine and/or sphingosine-1-phosphate prevent apoptosis of pancreatic β-cells and exert an anti-diabetic effect [30]. Sphingosine-1-phosphate binds to its receptors and activates AMPK, a serine/threonine kinase that stimulates glucose uptake and lipid oxidation [37,47,48].

Obesity induces low-grade inflammation that contributes to the development of obesity comorbidities, such as insulin resistance, T2DM, and atherosclerosis [49,50]. White adipose tissue (WAT) is not just a lipid storage organ but also an endocrine organ that affects metabolism and inflammation via the secretion of different cytokines and adipokines. WAT is composed of adipocyte precursors, adipocytes, and immune cells, mainly macrophages [51]. Adipocytes and macrophages that infiltrate adipose tissue secrete proinflammatory cytokines such as IL-1, IL-6, and TNF-α capable of impairing insulin signaling [51]. In obesity, adipocytes can increase either in size (hypertrophy) or in number (hyperplasia). Hypertrophy is associated with hypoxia of adipocytes, which activates the HIF-1 gene, leads to cell stress, and activates cell death and transcription factor NF-κB [19,50]. NF-κB stimulates the production of proinflammatory cytokines and chemokine monocytechemotactic protein-1 (MCP-1), which leads to macrophage recruitment and increased secretion of proinflammatory cytokines [19,50]. This significantly impairs the transduction of insulin signals in adipocytes and causes excessive lipolysis, cell death, and progression of insulin resistance [19]. In obese individuals the plasma levels of leptin are increased, while adiponectin levels are decreased [19,51]. The reduced circulating adiponectin levels in obesity result in reduced stimulation of AdipoRs and decreased ceramide degradation in tissues, which contributes to the development of insulin resistance [8].

## 4. Ceramides and Type 2 Diabetes

Lipotoxicity interferes with insulin signaling and contributes to the development of numerous diseases, including T2DM. Various lipids have been measured in diabetic patients, and ceramides have been proposed to play an important role in the pathogenesis of diabetes and its associated complications. Previous studies report that individuals with T2DM had higher serum and tissue ceramide levels than their nondiabetic counterparts and that ceramide levels were predictors of insulin resistance in T2DM [52,53]. The plasma concentration of dihydroceramide was elevated years prior to diabetes diagnosis and was associated with disease predisposition [54]. Previous studies report that insulin resistance correlated with ceramide levels in plasma, adipose tissue, liver, and skeletal muscles [55,56]. Research suggests that an accumulation of ceramides in the tissues impairs insulin signaling and glucose uptake by the PI3K pathway, which is the major pathway involved in glucose transport. Ceramides cause inactivation of PKB (Akt), which is needed for insulin signaling, and impair the translocation of the glucose transporter GLUT 4 to the cell membrane. Firstly, ceramides inhibit PKB through activation of its dephosphorylation, catalyzed by protein phosphatase 2A. Secondly, they inactivate PKB through its binding to the inhibitory PKCzeta protein. Ceramides may also decrease GLUT 4 gene transcription [12,57,58]. Therefore, ceramides prevent the uptake of glucose and its storage as glycogen and triglycerides. This results in elevated levels of glucose in the blood and decreased insulin sensitivity [57]. Previous studies show that circulating ceramides may be early diagnostic indicators of metabolic diseases like diabetes. Haus et al. showed that type 2 diabetic subjects had higher plasma concentrations of Cer-C18:0, C20:0, and C24:1 compared to healthy controls and that insulin sensitivity was inversely correlated with these ceramide levels [55]. Brozinick et al. revealed that distinct ceramide species with medium and long fatty acid chains were high in non-human primates (macaque monkeys) with prediabetes and diabetes [59]. Sokolowska et al. found that in adult patients with newly diagnosed diabetes, Cer-C22:0 and Cer-C24:0 positively correlated with total cholesterol, triglycerides, and low-density lipoprotein cholesterol (LDL), particularly in autoimmune diabetes, while in T2DM, there was a positive correlation only between Cer-C24:0 and total cholesterol. The authors conclude that monitoring Cer and sphingomyelins levels may help with more accurate differentiation between autoimmune diabetes and T2DM [60].

## 5. Ceramides and Cardiovascular Diseases 

Cardiovascular diseases remain the most common cause of death worldwide despite significant advancements in their therapy. In 2021 alone, 20.5 million people died from CVDs, representing approximately one-third of all global deaths [61,62,63]. The main contributors to CVDs include smoking, alcohol intake, hypertension, diabetes mellitus, obesity, and dyslipidemia [64,65].

The lipid profile is used as a predictor of CVD risk. The traditional blood tests of lipid status include LDL-cholesterol, HDL-cholesterol, and triglyceride levels, but these routine biomarkers fail to identify all patients at high risk of cardiovascular events [66,67]. Recent studies have identified ceramides to be involved in the development of CVDs, including atherosclerosis, coronary artery disease, heart failure, and myocardial infarction [10,12,68,69,70]. Dysregulation of ceramide and sphingolipid metabolism is suggested to be a key factor in cardiac lipotoxicity, and data from both animal and human studies show that sphingolipid accumulation in cardiomyocytes is associated with cardiac hypertrophy [71,72]. Ceramide has been considered a new predictor for CV events, with higher predictive value than LDL cholesterol [13,62,66]. It is suggested that Cers are involved in atherogenesis by promoting LDL aggregation, remodeling the vascular wall, and increasing inflammation [13,70]. One of the suggested mechanisms for ceramide-induced cardiovascular manifestations is that sphingomyelin accumulates in atherosclerotic lesions, where it may be hydrolyzed by SMases, thus increasing LDL-ceramide and their faster aggregation in the arterial wall, leading to atherosclerosis [42].

Ceramides were reported to activate TNF-α and NF-κB pathways, initiating an inflammation cascade. Plasma Cers were also found to be strongly correlated with the pro-inflammatory cytokine IL-6. Eventually, Cer-induced inflammation is a key factor for the development of atherosclerosis and CVDs [12,42].

Slijkhuis et al. investigated the lipid content of human carotid atherosclerotic plaque in detail and provided new insights into the role of lipids in atherosclerosis [73]. The authors identified SM and Cers as the lipid species that accumulate prominently in the calcified areas of the plaque. Other authors suggest that Cers have a crucial role in the pathogenesis of CVDs and may serve as new diagnostic markers for the progression of atherosclerosis [74]. Recent studies have reported an association between ceramides and cardiovascular risk factors such as age, arterial hypertension, and obesity and the altered ceramide profile observed with atherosclerosis progression. Certain types of ceramides are identified as proatherogenic mediators and are associated with increased cardiovascular risk [61,70,75,76]. Therefore, measurement of plasma ceramides may improve the identification of patients at high risk for cardiovascular events [62,77,78]. Ceramide risk score, composed of specific ceramides and their corresponding ratios, has been proposed as a promising indicator that provides additional predictive value for the assessment of cardiovascular disease risk in different populations [79,80]. The ceramide-based coronary event risk test (CERT) is determined based on four ceramides and their ratios, increasing with risk level. Ceramide score 1 (CERT 1), composed of Cer (d18:1/16:0), Cer (d18:1/18:0), Cer (d18:1/24:1), and their ratios to Cer (d18:1/24:0), applies a scale of 0 to 12 points, and patients are categorized into four risk groups (low–moderate–increased–high). The low-risk score is 0 to 2, the moderate-risk score is 3 to 6, the increased-risk score is 7 to 9, and high-risk is 10 to 12 [31,67,81]. Ceramides C16:0 and C18:0 are considered harmful, whereas those containing very long chains (e.g., C24) are considered protective. In order to improve the reliability of CERT 1, another ceramide risk score, i.e., CERT 2, has been proposed that is composed of four components—one ceramide/ceramide ratio, two ceramide/phosphatidylcholine (PC) ratios, and a single PC [67]. CERT 2 is a novel updated risk score able to predict major adverse cardiovascular events (MACEs) and cardiovascular diseases. Ceramides and phosphatidylcholines are incorporated into this parameter, which reflects the risk of CVD incidents [76,79]. CERT 1 and CERT 2 have been reported as stronger predictors for the assessment of CV risk compared to traditional biomarkers such as LDL cholesterol and high-sensitivity C-reactive protein. Both scores are effective tools able to predict CV events such as myocardial infarction, stroke, and cardiovascular death and could improve the identification of high-risk patients [11,31,67,76,79,80,82,83,84,85]. Previous studies have shown that blocking ceramide production can be an effective strategy for the treatment of CVD, and pharmacological inhibition of key enzymes and receptors in the three different processes of ceramides biosynthesis can reduce ceramide levels [62,75]. Clinical studies have confirmed that cholesterol-biosynthesis inhibitors, such as statins (which inhibit HMG-CoA reductase), effectively reduce plasma ceramides. Furthermore, proprotein convertase subtilisin/kexin type 9 (PCSK-9) inhibitors, monoclonal antibodies used in the therapy of hypercholesterolemia, and its related cardiovascular diseases were found to reduce both cholesterol and, potentially, ceramide levels and were suggested as a new therapeutic option for the treatment of dyslipidemia [62,67]. On the other hand, glucagon-like peptide 1 (GLP-1) receptor agonists (GLP-1 analogs), a class of medications used as therapeutic agents in T2DM, have shown cardioprotective effects against ceramide accumulation. GLP-1 analogs such as liraglutide lower glucose levels by stimulating pancreatic β-cells to release insulin. These agents also suppress the release of glucagon from α-cells and the hepatic glucose output, promote β-cell proliferation and reduce β-cell apoptosis, reduce appetite, and delay gastric emptying, thus contributing to long-term weight loss [12,86]. Although direct evidence is not yet available, it is suggested that a weight loss of at least 10% is associated with a 21% reduction in cardiovascular events [87]. According to the 2023 American Diabetes Association (ADA) guidelines, GLP-1 receptor agonists are mmended as a novel approach for mitigating cardiovascular risk. It is also reported that liraglutide inhibits the accumulation of C16:0 and C24:0 ceramide in mice liver, preventing subsequent inflammation and fibrosis [12]. Additional therapies targeting ceramide biosynthesis (e.g., myriocin—an inhibitor of serine palmitoyltransferase (SPT), the initial and rate-limiting enzyme in de novo synthesis of ceramide) and ceramide inhibitors (e.g., fumonisin B1, fungin FTY720, etc.), are under investigation [11,12]. Myriocin, a natural specific inhibitor of SPT, has shown promising results in the treatment of several diseases, including CVDs. It was reported to reduce atherosclerosis, hepatic steatosis, and fibrosis in mice induced by elevated ceramide levels. Moreover, it could restore normal vasodilation of blood vessels via activation of endothelial NO synthase (eNOS) and NO release [12]. However, future laboratory and clinical research on the therapeutic options for ceramide reduction is necessary. Traditional therapies such as lipid-lowering drugs, lifestyle changes, increased physical activity, and dietary interventions may be employed to control ceramide levels while specific ceramide- targeted drugs are being developed [31].

## 6. Ceramides and Inflammation

Inflammation is part of the body’s response mechanism to infection or tissue injury. It is the process by which the immune system recognizes and destroys harmful and foreign substances and begins the healing of tissues. In chronic conditions, persistent low-grade inflammation is often observed [13,88]. Elevated Cers lead to stimulated synthesis of proinflammatory mediators such as TNF-α, interferon-γ, IL-1β, or platelet-activating factor that enhance the activity of both lysosomal and membrane sphingomyelinases, thus stimulating Cer synthesis and incremental inflammation [11,13,30,40,89]. Ceramide production induces inflammation through activation of the proinflammatory transcription factor NF-κB. NF-κB is a family of transcription factors that induce genes encoding proinflammatory cytokines and proteins, such as IL-1β, IL-6, IL-8, monocyte chemoattractant protein-1, and COX-2, all of which play crucial roles in the inflammatory responses [90]. One of the key modulators of ceramide biosynthesis is Toll-like receptor-4 (TLR4). The receptor is activated on macrophages in response to bacterial infection and tissue damage. Ligands of TLR4 include lipopolysaccharides, low-density lipoproteins, viral proteins, polysaccharides, etc. Previous studies suggest that TLR signaling in macrophages is not only required for innate immunity against pathogens but also contributes to the pathogenesis of many diseases. One of the TLR4 signaling pathways leads to MAP kinases and NF-κB activation and the release of inflammatory molecules. In addition to cytokine production, another response of TLR4-mediated macrophage activation is the imbalance of cellular lipids. Recent studies have revealed that TLR4 agonists induce the expression of several enzymes in the de novo pathway of ceramide biosynthesis, which results in an increase in cellular sphingolipids, including ceramide [40,91,92,93,94].

## 7. Conclusions

Recent studies reveal the impact of ceramides on metabolic and cardiovascular health and provide evidence that the accumulation of ceramides induces cellular dysfunction. Large cohort human studies convincingly demonstrate that the simultaneous measurement of ceramides together with routine biomarkers of lipid profile may improve the diagnostic accuracy of metabolic diseases like diabetes, and may be a promising diagnostic tool for better assessment of the progression of cardiovascular diseases and a potent predictor of CV events. Therefore, further research on specific ceramide species and their tissue and organ distribution will shed light on the molecular mechanisms of ceramide-induced lipotoxicity. In conclusion, thorough understanding of the dysregulation of ceramide metabolism will add information to clarify the pathogenesis of a variety of diseases, and will help to identify better therapeutic targets and to develop effective treatments.

## Figures and Tables

**Figure 1 diseases-12-00195-f001:**
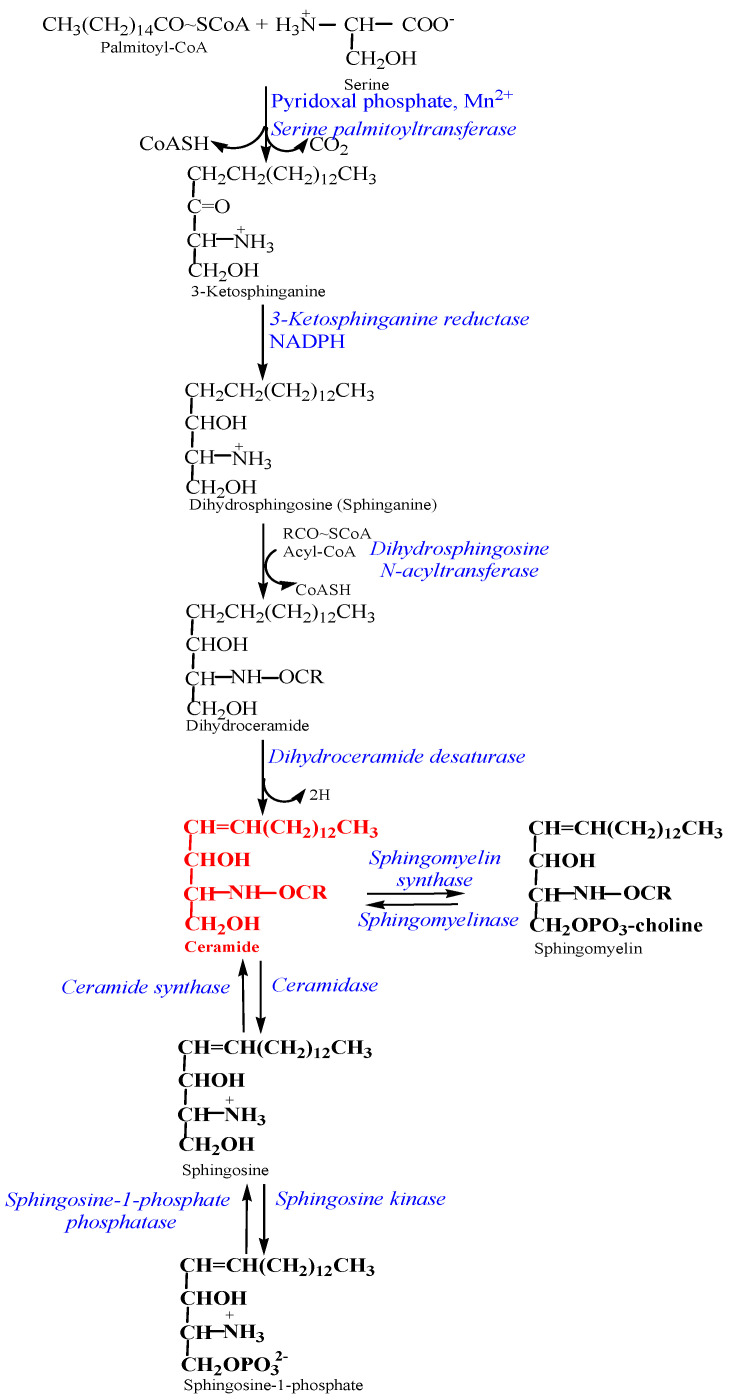
The three main pathways of ceramide biosynthesis. The de novo synthesis is a multistep pathway that starts with the condensation of serine with palmitoyl-CoA to form 3-ketosphinganine, which is the rate-limiting step and is catalyzed by serine palmitoyltransferase. In the second pathway, Cers are synthesized via the hydrolysis of the phosphocholine head groups of sphingomyelin by sphingomyelinases. In the salvage pathway, Cers are recycled from sphingosine re-acylation that is catalyzed by ceramide synthase.

**Figure 2 diseases-12-00195-f002:**
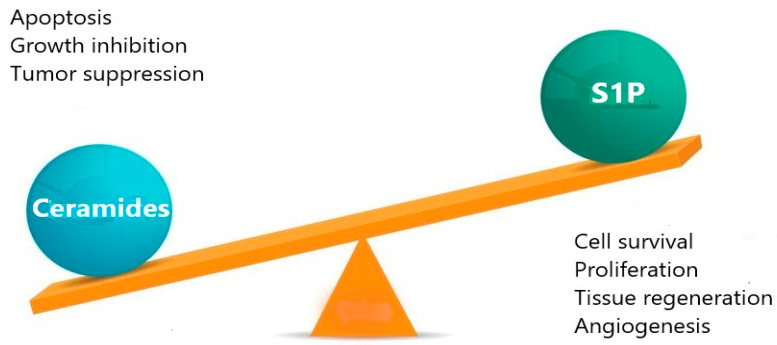
Sphingolipid rheostat and the opposite effects of ceramides and S1P on cell fate.

**Table 1 diseases-12-00195-t001:** Substrate specificity and tissue distribution of ceramide synthase isoforms.

Ceramide Synthase Isoforms	Acyl Chain-Length Specificity	Tissue Distribution
CerS1	C18	Muscle tissues, brain, testis, gastrointestinal tract, female tissues, bone marrow, and lymphoid tissues
CerS2	C22-24	Mainly expressed in liver, kidney, and small intestine
CerS3	Broad substrate specificity toward medium- to long-chain fatty acids	Endocrine tissues, gastrointestinal tract, male tissues, and female tissues, especially in esophagus, testis, and skin
CerS4	C20-C24	Nearly all tissues
CerS5	C16	All tissues except heart muscle and smooth muscle, especially in adipose tissue
CerS6	C16	Most tissues, especially in brain and adipose tissue

## Data Availability

Review, no original data sheets are available.

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
