# Peer review of "Ceramides—Emerging Biomarkers of Lipotoxicity in Obesity, Diabetes, Cardiovascular Diseases, and Inflammation"

_diseases, 2024, doi:10.3390/diseases12090195_

Round 1
Reviewer 1 Report
Comments and Suggestions for Authors
Overall this manuscript is weak, their is nothing intrinsically wrong with the information that is included however, it reads as a collection of points than a true review that should include analysis of the information presented, but that is missing here. As a result of this in its current form I think it would be of limited interest to readers, I would suggest including detailed analysis of the information provided.
Specific comments
I think that figure 1 would be clearer if the authors included the structures of the molecules rather than the text representation and the legend could be more descriptive.
In line 47 the authors state that Cer are elevated in the blood of obese patients but so are many other lipids, why s the increase in Cer so important?
Line 54-60 contains a nice description of the distribution of the ceramide synthases but discussion of the biological significance of this Should have been included.
Line 76 the author’s state that previous studies have shown that adiponectuin controls ceramide levels but no reference is provided
Line 88 the authors state that mild inflammation is present in obesity and that this helps drive obesity associated comorbidities but doesn’t really say how.
Line 100 the author states that adiponectin levels are decreased also leading to diabetes but again doesn’t explain how this happens.
Line 118 the authors state ‘primates’ what species?
Line 130 the authors state that ‘The standard measurement include LDL...’ I am not sure what the author is trying to say here.
Line150-155 I understand what this text is trying to say but I had to read it several times to understand it I would suggest trying to reword to improve clarity.
Line169 authors state ‘could significantly improve’ this is vague, please state the detail that supports this statement.
The conclusion is vague its doesn’t really tell the reader anything not already stated in the paper.
Comments on the Quality of English Language
The quality of the written English is fine.
Author Response
Dear reviewer,
Thank you for your critical and expert review. We have carefully analyzed and followed it . The initial version of the present manuscript that you have received was revised and significantly enriched upon the first submission according to the academic reviewer's recommendations. We believe that the revised manuscript is improved and will contribute to the field of biomarkers research and particularly lipidomics.
Responses to the specific comments:
1. I think that figure 1 would be clearer if the authors included the structures of the molecules rather than the text representation and the legend could be more descriptive.
Response: The missing structures are now added in Figure 1 and the legend is made more descriptive.
2. In line 47 the authors state that Cer are elevated in the blood of obese patients but so are many other lipids, why is the increase in Cer so important?
Response: In the last version of the manuscript this sentence is in line 94 of page 3 at the end of section 2 and is used as a link between Section 2 and the other sections of the review. It is true that various lipids increase in obesity including TAG, DAG, long-chain fatty acids, phospholipids, and ceramides. This is discussed in Section 3 of the manuscript - Ceramides and obesity.
3. Line 54-60 contains a nice description of the distribution of the ceramide synthases but discussion of the biological significance of this Should have been included.
Response: Information about the biological significance of ceramide synthases is added, line 81-84, page 2. A table with the substrate specificity and tissue distribution of the five ceramide synthase isoforms was also included (Table 1).
4. Line 76 the author’s state that previous studies have shown that adiponectuin controls ceramide levels but no reference is provided.
Response: The references are now added, line 134, page 5.
5. Line 88 the authors state that mild inflammation is present in obesity and that this helps drive obesity associated comorbidities but doesn’t really say how.
Response: In order to support this statement we add information about the function of white adipose tissue (WAT) as an endocrine organ, line 163-167, page 5.
As an endocrine organ WAT is the major culprit of the obesity-associated disorders. WAT secretes proinflammatory cytokines that trgger inflammatory processes and apoptosis. This state of low-grade chronic inflammation initiates lipotoxicity and systemic metabolic inflammation in obesity.
6. Line 100 the author states that adiponectin levels are decreased also leading to diabetes but again doesn’t explain how this happens.
Response: The reduced circulating adiponectin levels in obesity results in reduced stimulation of AdipoRs and decreased ceramide degradation in tissues which contributes to the development of insulin resistance (line 177-180, page 6).
In the next section of the review (Section 4) we explain in details the mechanisms by which ceramide accumulation leads to insulin resistance, line 191-200, page 6.
7. Line 118 the authors state ‘primates’ what species?
Response: The animals used in the cited study are non-human primates, macaque monkeys, line 205, page 6.
8. Line 130 the authors state that ‘The standard measurement include LDL...’ I am not sure what the author is trying to say here.
Response: We replaced "standard measurement" with "traditional blood tests of lipid status", line 219, page 7.
9. Line150-155 I understand what this text is trying to say but I had to read it several times to understand it I would suggest trying to reword to improve clarity.
Response: We reworded the text to make it clearer, line 248-252, page 7.
10. Line169 authors state ‘could significantly improve’ this is vague, please state the detail that supports this statement.
Response: We give more details to support the statement, line 264-269, page 7.
11. The conclusion is vague its doesn’t really tell the reader anything not already stated in the paper.
Response: The conclusion has been edited and enriched, the changes are highlighted.
Reviewer 2 Report
Comments and Suggestions for Authors
This review article deserves to be accepted for publication. The authors prepared a succinct review that addressed essential aspects of the role of ceramides in the pathogenesis of several diseases. It would be helpful to expand the text with the author's personal experience to avoid mere citation of literature data. Also, a paragraph on lipid signaling in the cell would improve the manuscript.
Author Response
1. It would be helpful to expand the text with the author's personal experience to avoid mere citation of literature data.
Response: Our personal experience on lipidomics is not associated with ceramides that are the focus of the present review. Therefore, we have not included own previous studies. We intend to expand the area of our future investigations with the analysis of lipid biomarkers that could be early indicators of cellular and metabolic dysfunction.
2. A paragraph on lipid signaling in the cell would improve the manuscript.
Response: Since the present review aims to explore ceramides and their detrimental accumulation in tissues, we added a paragraph how ceramide signaling leads to metabolic dysfunction , line 45-51, page 2.